Association between hypomagnesemia and mortality among dialysis patients: a systematic review and meta-analysis

Huang Chi-Ya 1
Yang Chi-Chen 1
Hung Kuo-Chuan 2
Jiang Ming-Yan 3
Huang Yun-Ting 3
Hwang Jyh-Chang 3 4
Hsieh Chih-Chieh 5
Chuang Min-Hsiang 1
Chen Jui-Yi 3 6 kwuilus0101@gmail.com
1 Department of Internal Medicine, Chi Mei Medical Center , Tainan , Taiwan
2 Department of Anesthesiology , Chi Mei Medical Center, Tainan , Taiwan
3 Division of Nephrology, Department of Internal Medicine, Chi Mei Medical Center , Tainan , Taiwan
4 Department of Hospital and Health Care Administration, Chia Nan University of Pharmacy and Science , Tainan , Taiwan
5 Division of Nephrology, Department of Internal Medicine, Pingtung Christian Hospital , Pingtung , Taiwan
6 Department of Health and Nutrition, Chia Nan University of Pharmacy and Science , Tainan , Taiwan
Capusa Cristina
Electronic publication date: 2022 Oct 11
Publication date: 2022
Volume: 10
Electronic Location ID: e14203
Received 2022 Apr 13; Accepted 2022 Sep 19
Copyright: © 2022 Huang et al.
Copyright year: 2022
Copyright holder: Huang et al.
License: This is an open access article distributed under the terms of the Creative Commons Attribution License, which permits unrestricted use, distribution, reproduction and adaptation in any medium and for any purpose provided that it is properly attributed. For attribution, the original author(s), title, publication source (PeerJ) and either DOI or URL of the article must be cited.
License URL: https://creativecommons.org/licenses/by/4.0/

Keywords: Hypomagnesemia, Magnesium, Dialysis, Mortality, Hemodialysis

Funding: Chi-Mei Medical Center CMFHR10973 This study was supported by Chi-Mei Medical Center (CMFHR10973). The funders had no role in study design, data collection and analysis, decision to publish, or preparation of the manuscript.

==============================
Background

Malnutrition-inflammation-atherosclerosis (MIA) syndrome is caused by the inflammatory cytokines in end stage renal disease (ESRD) patients, and MIA complex-related factors may be associated with hypomagnesemia and mortality. However, the association between serum magnesium level and mortality for dialysis patients is still not clear. Additionally, no meta-analysis has investigated the impact of serum magnesium on peritoneal dialysis and hemodialysis, separately.

Methods

We searched published studies in PubMed, Embase, Cochrane, Collaboration Central Register of Controlled Clinical Trials, and Cochrane Systematic Reviews through April 2022. Studies associated with serum magnesium and all-cause mortality or cardiovascular (CV) mortality in ESRD on kidney replacement therapy (KRT) patients were included. A hazard ratio (HR) with 95% confidence intervals (CI) was used to report the outcomes.

Results

Twenty-one studies involving 55,232 patients were included. Overall, there was a significant association between hypomagnesemia and all-cause mortality for dialysis patients (HR: 1.67, 95% CI [1.412–2.00], p < 0.001; certainty of evidence: moderate) using a mixed unadjusted and adjusted HR for analysis. There was also a significantly increased risk of CV mortality for individuals with hypomagnesemia compared with the non-hypomagnesemia group (HR 1.56, 95% CI [1.08–2.25], p < 0.001; certainty of evidence: moderate). In addition, a subgroup analysis demonstrated that hypomagnesemia was associated with a high risk of both all-cause mortality and CV mortality (all-cause mortality, HR:1.80, 95% CI [1.48–2.19]; CV mortality, HR:1.84, 95% CI [1.10–3.07]) in hemodialysis (HD) patients, but not in participants receiving peritoneal dialysis (PD; all-cause mortality, HR:1.26, 95% CI [0.84–1.91]; CV mortality, HR:0.66, 95% CI [0.22–2.00]). The systematic review protocol was prespecified and registered in PROSPERO [CRD42021256187].

Conclusions

Hypomagnesemia may be a significant risk factor for all-cause mortality and CV mortality in KRT patients, especially in those receiving hemodialysis. However, because of the limited certainty of evidence, more studies are required to investigate this association.

Introduction

Hypermagnesemia is common in patients with end stage renal disease (ESRD) due to dialysate with a magnesium content of 0.5 mmol/L (Yang et al., 2021). Patients receiving regular dialysis may have hypomagnesemia because increased phosphate, sulfate, or other anions would decrease serum magnesium (Mg) levels by binding Mg and forming Mg complex. In Japan, about 0.38% of patients on hemodialysis have a serum Mg level less than 2.5 mg/dL (Sakaguchi et al., 2014). Among dialysis patients, magnesium deficiency has been associated with inflammation, hyperparathyroidism, insulin resistance-related diabetes mellitus, oxidative stress, atherosclerosis, and calcification of vascular smooth muscle cells (VSMC), which may lead to hypertension (Salem et al., 2012; Montes de Oca et al., 2014). A previous meta-analysis concluded that hypomagnesemia is significantly associated with all-cause and cardiovascular mortality in populations with chronic kidney disease (CKD) and ESRD without analyzing the outcomes for these two populations separately (Xiong et al., 2019).

Therefore, we defined hypomagnesemia as a serum Mg level less than 2.5 mg/dL and conducted a meta-analysis to explore the associations among serum magnesium, all-cause mortality, and CV mortality in patients with ESRD on kidney replacement therapy (KRT).

Materials and Methods

Search strategy and selection criteria

This meta-analysis was conducted according to the Preferred Reporting Items of Systematic Reviews and Meta-Analyses (PRISMA) statement (Higgins et al., 2011) and following the Cochrane methods (Salguero et al., 2008). The protocol was prospectively submitted to PROSPERO [CRD42021256187].

Two investigators (CY Huang; CC Yang) searched published studies in PubMed, Embase, Cochrane, Collaboration Central Register of Controlled Clinical Trials, and Cochrane Systematic Reviews until April 2022 without any language limitation (Data S1). Using Medical Subject Headings (MeSH) terms and the PICO (population, intervention, comparison, outcome), we selected key words such as “Renal Dialysis,” “Hemodialysis,” “Peritoneal dialysis,” “Magnesium,” “Hypomagnesemia,” “Hypermagnesemia,” “Mortality,” and “Death” as search terms for our literature review. Prospective and retrospective cohort studies and observational studies were included, but case reports and case series were excluded. Two investigators (CY Huang, CC Yang) independently performed searches and checked all articles for inclusion. If they disagreed on the inclusion of an article, a third author (MY, Jiang) resolved the dispute.

Inclusion and exclusion criteria

Published studies were included if they: (1) reported the impact of serum magnesium on all-cause or CV mortality for KRT patients; (2) used a cohort study design, including retrospective (e.g., cohort and case control study) and prospective cohort studies; and (3) reported the hazard ratio (HR) with 95% confidence intervals (CIs) or provided sufficient data to investigate the outcomes. The following studies were excluded: (1) studies that did not include the outcomes of CV mortality or all-cause mortality, or (2) had follow-up times less than 12 months.

Data extraction

The following data were extracted from the articles: first author, year of publication, patient characteristics (sample size, age, and sex), follow-up duration, and clinical outcomes. Both abstracts and full papers were selected for quality assessment and data synthesis. If the data were incomplete in the text but extractable from the figures (ex. Kaplan-Meier (KM) survival curve), we used WebPlotDigitizer Version 4.1 to extrapolate the data (Drevon, Fursa & Malcolm, 2017). Another author (YT, Huang) then verified these data. We did our best to contact the corresponding authors for more information about the selected studies.

Quality assessment

The risk of bias of the included studies was assessed by two authors (CC, Hsieh; MH, Chuang), independently, using the Newcastle-Ottawa Scale (NOS). The NOS contains three domains which consisted of eight items totally, which represent three quality assessment parameters: selection, comparability, and outcome. One point was given for a low risk of bias, and 0 points were given for a high or unclear risk of bias. A score of 0–3 points was considered to be a low-quality study, a score of 4–6 points equated to a moderate-quality study, with a score of 7–9 points indicating a study was high quality.

Outcomes and definition of hypomagnesemia or non-hypomagnesemia

The primary outcome of this meta-analysis was all-cause mortality, with CV mortality as a secondary outcome. Individual trials had different serum magnesium cut-off values for defining hypomagnesemia and non-hypomagnesemia in KRT patients. Among these studies, the HRs of serum magnesium as dichotomous variables were regarded as “lower magnesium vs. non-hypomagnesium,” according to their respective categories.

Data synthesis and statistical analysis

The random-effects model was employed to analyze the selected outcomes (all-cause mortality and CV mortality) between the two groups. The effect size was expressed as the pool HRs with 95% CIs.

We used funnel plots to demonstrate potential publication bias. Statistical heterogeneity was calculated using I2, with an I2 more than 50% indicating the studies were not homogeneous. P-values < 0.05 were considered statistically significant. We used the Grading of Recommendations Assessment, Development, and Evaluation (GRADE) approach to assess the certainty of the evidence for each outcome. We used the Comprehensive Meta-Analysis software (Version 3.3.070, November 20, 2014) for all statistical analyses.

Subgroup analysis

We performed subgroup analyses of adjusted/unadjusted outcomes, dialysis modality (HD, PD), serum magnesium cut-off value for defining the hypomagnesemia and non-hypomagnesemia groups (1.58–2.5 mg/dL vs. 2.5–2.79 mg/dL), follow-up period (≥5 years vs. <5 years), study design (retrospective vs. prospective), and if the data were extracted from the KM survival curve.

Results

Search results and included studies

Our initial screening of cohort studies focused on published studies that were available on May 20th, 2021. We then submitted our study to PROSPERO and completed our comprehensive database collection on April 4th, 2022.

A total of 3,416 studies were identified from the databases and 224 studies were identified from registers as original research. We removed 810 studies due to duplication. Of the remaining 2,830 articles, a further 2,806 articles were excluded after reviewing the title and abstract (Fig. 1). The remaining 24 studies were then assessed for eligibility. After excluding three more studies that lacked usable data for an analysis, a total of 21 observational studies of 55,232 patients undergoing dialysis were selected for inclusion in the final meta-analysis.

Figure 1 Flowchart of study selection for meta-analysis.

Study characteristics

As shown in Table 1, the included studies were published from 2007 to 2020, including 12 studies from Asia (Ishimura et al., 2007; Kurita et al., 2015; Ago et al., 2016; Cai et al., 2016; Yang et al., 2016; Ye et al., 2018; Mizuiri et al., 2019; Shimohata et al., 2019; Tamura et al., 2019; Wu et al., 2019; Lu et al., 2020; Mizuiri et al., 2020), seven from Europe (Markaki et al., 2012; Vervloet et al., 2013; João Matias et al., 2014; Lacson et al., 2015; de Roij van Zuijdewijn et al., 2015; Garagarza et al., 2015; Selim et al., 2017), and two from North America (Fein et al., 2014; Li et al., 2015). Thirteen studies were retrospective studies (Ishimura et al., 2007; Fein et al., 2014; de Roij van Zuijdewijn et al., 2015; Lacson et al., 2015; Ago et al., 2016; Cai et al., 2016; Li et al., 2016; Yang et al., 2016; Mizuiri et al., 2019; Shimohata et al., 2019; Wu et al., 2019; Lu et al., 2020; Mizuiri et al., 2020), and the other eight studies were prospective studies (Markaki et al., 2012; Vervloet et al., 2013; João Matias et al., 2014; Garagarza et al., 2015; Kurita et al., 2015; Selim et al., 2017; Ye et al., 2018; Tamura et al., 2019).

Table 1 Summary of the baseline characteristics of the included studies.

Author	Nation	Study design	Modality	Population (n)	Male (%)	DM (%)	*Age (years)	Follow up duration (m)	Cut-off value of serum Mg (mg/dL)	
Ishimura et al. (2007)	Japan	Retrospective	HD	515	59.4	24.0	60 ± 12	51	2.77	
Markaki et al. (2012)	Greece	Prospective	HD+PD	74	55.4	18.9	65 ± 15	50	2.45	
Vervloet et al. (2013)	Germany	Prospective	HD+PD	761	59.0	25.0	63 ± 14	36	2.07	
João Matias et al. (2014)	Portugal	Prospective	HD	206	55.0	26.0	63.6 ± 14.3	48	2.79	
Fein et al. (2014)	United States	Retrospective	PD	62	45.0	25.0	55 ± 16	129.6	1.94	
Lacson et al. (2015)	Germany	Retrospective	HD	27,544	53.7	53.6	61.9 ± 15	12	1.58	
de Roij van Zuijdewijn et al. (2015)	Netherlands	Retrospective	HD	365	61.9	20.8	64.1 ± 13.7	36	2.07	
Garagarza et al. (2015)	Portugal	Prospective	HD	605	NR	NR	69.9	NR	2.0	
Kurita et al. (2015)	Japan	Prospective	HD	2,185	62.0	26.0	61.7 ± 12.5	36	2.3	
Li et al. (2015)	United States	Retrospective	HD	9,359	56.2	59.1	63.3 ± 14.9	60	2.0	
Yang et al. (2016)	China	Retrospective	PD	10,692	55.0	63.0	56 ± 16	13	1.8	
Cai et al. (2016)	China	Retrospective	PD	253	55.3	22.9	58 ± 16	29	1.7	
Ago et al. (2016)	Japan	Retrospective	HD	399	63.2	35.3	65.86 ± 11.8	12	2.2	
Selim et al. (2017)	Republic of Macedonia	Prospective	HD	185	59.5	17.3	49.7 ± 14.7	60	2.67	
Ye et al. (2018)	China	Prospective	PD	402	57.0	20.6	49.3 ± 14.9	49.9	1.7	
Wu et al. (2019)	China	Retrospective	HD	169	53.8	NR	60.20 ± 15.64	120	2.43	
Tamura et al. (2019)	Japan	Prospective	HD	392	65.3	47.2	68	43.5	2.6	
Mizuiri et al. (2019)	Japan	Retrospective	HD	353	66.6	40.2	68	36	2.4	
Shimohata et al. (2019)	Japan	Retrospective	HD	83	62.1	0	58	120	2.5	
Mizuiri et al. (2020)	Japan	Retrospective	HD	215	67.9	44.2	73	36	2.3	
Lu et al. (2020)	China	Retrospective	HD	413	57.4	14.4	50.4 ± 14.3	12	2.43	
Notes:

* Data of age are presented as mean ± standard deviation.

Abbreviations: DM, Diabetes Mellitus; HD, Hemodialysis; m, month; Mg, Magnesium; mg/dl, milligrams per deciliter; NR, not reported; PD, Peritoneal dialysis.

Hemodialysis (HD) patients were the study population in 15 studies, (Ishimura et al., 2007; João Matias et al., 2014; de Roij van Zuijdewijn et al., 2015; Garagarza et al., 2015; Kurita et al., 2015; Lacson et al., 2015; Li et al., 2015; Ago et al., 2016; Selim et al., 2017; Mizuiri et al., 2019; Shimohata et al., 2019; Tamura et al., 2019; Wu et al., 2019; Lu et al., 2020; Mizuiri et al., 2020), peritoneal dialysis (PD) patients were the study population in four studies (Fein et al., 2014; Cai et al., 2016; Yang et al., 2016; Ye et al., 2018), and two studies had a study population that combined HD and PD patients (Markaki et al., 2012; Vervloet et al., 2013). The mean ages of the HD and PD populations were 62 ± 6 and 55 ± 6 years, respectively. The follow-up duration of the 21 included studies ranged from 12 to 129.6 months, with one study not reporting the follow-up duration (Garagarza et al., 2015). Twenty studies reported the association of serum magnesium and all-cause mortality (Ishimura et al., 2007; Markaki et al., 2012; Fein et al., 2014; João Matias et al., 2014; de Roij van Zuijdewijn et al., 2015; Garagarza et al., 2015; Kurita et al., 2015; Lacson et al., 2015; Li et al., 2015; Ago et al., 2016; Cai et al., 2016; Yang et al., 2016; Selim et al., 2017; Ye et al., 2018; Mizuiri et al., 2019; Shimohata et al., 2019; Tamura et al., 2019; Wu et al., 2019; Lu et al., 2020; Mizuiri et al., 2020) and nine studies reported the association of serum magnesium and CV mortality (Ishimura et al., 2007; Vervloet et al., 2013; João Matias et al., 2014; de Roij van Zuijdewijn et al., 2015; Cai et al., 2016; Ye et al., 2018; Mizuiri et al., 2019; Tamura et al., 2019; Lu et al., 2020). The cut-off value of serum magnesium used for defining hypomagnesemia ranged from 1.7 to 2.77 mg/dl.

Heterogeneity and publication bias

The heterogeneity was 82.77% for all-cause mortality and 71.46% for CV mortality according to the I2 test. Publication bias, as evaluated using funnel plots and Egger’s test, was significant for all-cause mortality (Fig. S1), but not significant for CV mortality (Fig. S2). We used subgroup analyses and meta-regression to investigate the observed heterogeneity.

Quality of enrolled trials

The NOS score of the included studies was 6–9 (Table S1), with 20 of the 21 studies reaching the high-quality NOS score range of 7–9. The one study that did not reach a score between 7–9 (Garagarza et al., 2015) received a score of six points. This study was considered to be of moderate quality because it did not include the follow-up duration of the study. There was a high inter-observer reliability in article selection (Kappa coefficient = 0.90).

Assessment of evidence quality and summary of findings

An evidence quality assessment was performed using the GRADE system for the outcomes of all-cause mortality and CV mortality (Table S3).

Serum magnesium and all-cause mortality

Twenty studies of 54,471 KRT patients reported the association between serum magnesium levels and all-cause mortality. Ten studies presented both unadjusted and adjusted HR (de Roij van Zuijdewijn et al., 2015; Kurita et al., 2015; Li et al., 2015; Ago et al., 2016; Cai et al., 2016; Yang et al., 2016; Selim et al., 2017; Shimohata et al., 2019; Wu et al., 2019; Mizuiri et al., 2020), seven studies included only adjusted HR (Ishimura et al., 2007; Markaki et al., 2012; Fein et al., 2014; João Matias et al., 2014; Garagarza et al., 2015; Ye et al., 2018; Lu et al., 2020), and three studies included only unadjusted HR (Lacson et al., 2015; Mizuiri et al., 2019; Tamura et al., 2019). Four studies extracted data for the HR calculation from the KM survival curve (Table 2; Fein et al., 2014; Mizuiri et al., 2019; Tamura et al., 2019; Lu et al., 2020; Mizuiri et al., 2020). The results of our meta-analysis showed that hypomagnesemia was associated with increased risk of all-cause mortality compared with non-hypomagnesemia among KRT patients (random effect, HR:1.67, 95% CI [1.41–2.00], p < 0.001, (Fig. 2A)). However, we found high heterogeneity among the included studies (random effect model, I2 value of 82.77%, Fig. 2A).

Table 2 Summary of the outcome of the included studies.

Author	All-cause mortality, HR (95% CI)	CV mortality, HR (95% CI)	Primary outcome	Secondary outcome	Study quality, NOS	
Ishimura et al. (2007)	2.060a [1.026–4.135]	1.020a [0.323–3.220]	all-cause mortality	CV mortality	9	
Markaki et al. (2012)	1.160a [0.340–3.959]	NR	all-cause mortality	NR	7	
Vervloet et al. (2013)	NR	1.560a [0.950–2.561]	CV mortality	NR	7	
João Matias et al. (2014)	1.149a [1.005–1.314]	1.220a [0.732–2.033]	all-cause and CV mortality	NR	8	
Fein et al. (2014)	2.550b [1.398–4.652]	NR	all-cause mortality	NR	9	
Lacson et al. (2015)	1.600 [1.303–1.965]	NR	all-cause mortality	NR	8	
de Roij van Zuijdewijn et al. (2015)	1.140 [1.013–1.283]	1.370a [1.173–1.600]	all-cause and CV mortality	NR	9	
Garagarza et al. (2015)	2.040a [1.508–2.759]	NR	all-cause mortality	NR	6	
Kurita et al. (2015)	1.730 [1.201–2.492]	NR	all-cause mortality	NR	7	
Li et al. (2015)	1.170a [1.051–1.302]	NR	all-cause mortality	NR	9	
Yang et al. (2016)	0.970a [0.814–1.156]	NR	all-cause mortality	NR	9	
Cai et al. (2016)	0.075a [0.010–0.557]	0.013a [0.001–0.162]	all-cause and CV mortality	NR	9	
Ago et al. (2016)	2.410a [1.461–3.975]	NR	all-cause mortality	NR	7	
Selim et al. (2017)	1.140a [0.445–2.922]	NR	all-cause mortality	NR	8	
Ye et al. (2018)	1.530a [0.980–2.389]	1.430a [0.799–2.558]	CV mortality	All-cause mortality	8	
Wu et al. (2019)	8.300a [4.258–16.181]	NR	all-cause mortality	NR	9	
Tamura et al. (2019)	1.550b [1.040–2.310]	1.698b [0.877–3.289]	all-cause mortality	CV mortality	8	
Mizuiri et al. (2019)	2.790 [1.831–4.250]	3.730 [2.031–6.849]	all-cause and CV mortality	NR	8	
Shimohata et al. (2019)	2.730a [1.072–6.953]	NR	all-cause mortality	NR	7	
Mizuiri et al. (2020)	1.720b [1.007–2.939]	NR	all-cause mortality	NR	8	
Lu et al. (2020)	3.535 [1.594–7.838]	4.285 [1.422–12.910]	all-cause and CV mortality	NR	7	
Notes:

a Adjust HR.

b Data extracted from Kaplan-Meier survival curve.

Abbreviations: CV, cardiovascular; HR, hazard ratio; NOS, Newcastle-Ottawa Scale; NR, Not reported.

Figure 2 Forest plot showing increased risk of (A) all-cause mortality (B) cardiovascular mortality, comparing hypoMg vs. non-hypoMg in a population dialysis patients.

Serum magnesium and cardiovascular mortality

Nine out of the 21 studies reported the relationship between serum magnesium level and CV mortality among dialysis patients (Ishimura et al., 2007; Vervloet et al., 2013; João Matias et al., 2014; de Roij van Zuijdewijn et al., 2015; Cai et al., 2016; Ye et al., 2018; Mizuiri et al., 2019; Tamura et al., 2019; Lu et al., 2020). Four of these studies presented both unadjusted and adjusted HR (João Matias et al., 2014; de Roij van Zuijdewijn et al., 2015; Cai et al., 2016; Ye et al., 2018), three included only unadjusted HR (Tamura et al., 2019; Lu et al., 2020; Mizuiri et al., 2020), and the remaining two studies only included adjusted HR (Ishimura et al., 2007; Vervloet et al., 2013). One of these studies extracted data for the HR calculation from the KM survival curve. The results of this meta-analysis showed an increased risk of CV mortality for dialysis patients with hypomagnesemia compared to those with non-hypomagnesemia. (random effect, HR:1.56, 95% CI [1.08–2.25], p < 0.001, (Fig. 2B)). However, we found high heterogeneity among the included studies (random effect model, I2 value of 71.46%, Fig. 2B).

Subgroup analysis

A subgroup analysis showed that the association between hypomagnesemia and increased all-cause mortality (Fig. 3A) and CV mortality (Fig. 3B) among dialysis patients was consistently significant when stratified by follow-up duration and data extraction. A subgroup analysis by dialysis modality revealed that hypomagnesemia was associated with increased all-cause and CV mortality among HD patients, but this association was not present among PD patients. In addition, while the associations between hypomagnesemia and all-cause mortality were significant for serum magnesium cut-off values both above and below 2.5 mg/dL, the association between hypomagnesemia and CV mortality was only significant for a cut-off value less than 2.5 mg/dL.

Figure 3 Subgroup analysis for (A) all-cause mortality (B) cardiovascular mortality comparing hypoMg vs. non-hypoMg in a population of dialysis patients.

Meta-regression analysis

The quantitative measures of serum magnesium levels, older age, prevalence of diabetes, or male proportion were not associated with all-cause mortality (Age, Z = 0.12, p = 0.90, Fig. S3A; DM, Z = −1.27; p = 0.205, Fig. S3B; Male, Z = 0.35, p = 0.72, Fig. S3C; the quantitative measures of serum levels of Mg, Z = 0.95, p = 0.34) or CV mortality (Age, Z = 0.29, p = 0.7685, Fig. S4A; DM, Z = 0.75; p = 0.4548, Fig. S4B; Male, Z = 1.69, p = 0.0.0907, Fig. S4C; the quantitative measures of serum levels of Mg, Z = 0.91, p = 0.3636) between the hypomagnesemia and non-hypomagnesemia groups (Table S4).

Discussion

In this meta-analysis of twenty-one studies, our results showed that hypomagnesemia is associated with an increased risk of all-cause mortality and CV mortality among KRT patients. When stratified by dialysis modality, hypomagnesemia was associated with increased all-cause and CV mortality among HD patients, but this association was not significant among PD patients. Additionally, if the cut-off value of serum magnesium was higher than 2.5 mg/dl, hypomagnesemia did not significantly correlate with an increased risk of CV mortality among dialysis patients. Several mechanisms could explain the increased risk of all-cause mortality/CV mortality with serum magnesium deficiency. First, hypomagnesemia is associated with, and may lead to metabolic syndrome and insulin resistance (Volpe, 2008). Because magnesium is involved in over 300 enzymatic reactions, magnesium deficiency can contribute to the defective tyrosine kinase activities of insulin receptors (Suárez et al., 1995), impairment of modulating insulin-mediated glucose uptake and vascular tone (Barbagallo et al., 2003), elevation of tumor necrosis factor-alpha (Rodriguez-Morán & Guerrero-Romero, 2004), and CRP levels (Dibaba, Xun & He, 2014). Additionally, ESRD patients in the presence of hypomagnesemia have elevated levels of adiponectin, which plays a role in energy homeostasis and lipid/glucose metabolism (Markaki et al., 2012) and may correlate with ischemic heart disease (van de Wal-Visscher, Kooman & van der Sande, 2018). Hypomagnesemia is also related to defect immunity, involving both innate and acquired immune responses (Tam et al., 2003). The molecular identification of multiple magnesium transporters has provided the basis for the role magnesium plays in the immune system (Brandao et al., 2013). A low magnesium level inhibits endothelial proliferation by generating an inflammatory, pro-thrombotic, pro-atherogenic environment, which could be a contributor to the pathogenesis of cardiovascular disease (Maier et al., 2004). Patients with hypomagnesemia may have a higher risk of electrolyte imbalances such as hypokalemia or hypocalcemia, because dialysis patients with imbalances and an inadequate protein and energy intake develop protein-energy wasting (PEW; Maraj et al., 2018). Acute nutritional depletion seems to be related to poor appetite with low food intake and may be the cause of hypokalemia and hypocalcemia in dialysis patients (Vavruk et al., 2012). These electrolyte imbalances can cause cardiac arrhythmias (Hansen & Bruserud, 2018). A diet high in green leafy vegetables, cereals, and nuts abundant in magnesium might help prevent hypomagnesemia-related mortality in dialysis patients (Chrysant & Chrysant, 2019). Closely tracking serum magnesium levels is also an important strategy for preventing hypomagnesemia in dialysis patients.

In this meta-analysis, hypomagnesemia had a significant association with all-cause mortality/CV mortality for HD patients, but no obvious association was observed in PD patients in a subgroup analysis (Markaki et al., 2012; Fein et al., 2014; Cai et al., 2016; Yang et al., 2016). This could be because hemodynamic fluctuation happens in the intermittent delivery of HD due to the rapid loss of residual renal function and HD-induced myocardial and cerebral ischemia (Selby & Kazmi, 2019).

Another subgroup analysis revealed that while the associations between hypomagnesemia and all-cause mortality were significant for serum magnesium cut-off values both above and below 2.5 mg/dL, the association between hypomagnesemia and CV mortality was only significant for a cut-off value less than 2.5 mg/dL. Courivaud & Davenport (2014) reported that mild hypermagnesemia may have a protective effect in reducing soft tissue and vascular calcification, while higher serum magnesium may be the consequence of better nutrition status (Joffres, Reed & Yano, 1987). These findings may help explain why a lower serum magnesium cut-off value (≤2.5 mg/dl) was associated with a higher risk of CV mortality when a lower cut-off value (>2.5 mg/dl) was not (Courivaud & Davenport, 2014).

According to the 2020 United States Renal Data System Annual Data Report: Epidemiology of Kidney Disease, individuals with end stage renal disease (ESRD) undergoing dialysis have a higher risk of cardiovascular disease and a shorter life expectancy than the general population. The traditional risk factors of cardiovascular disease among dialysis patients include diabetes, hypertension, and smoking, while malnutrition, chronic inflammation, and vascular calcification are considered non-traditional risk factors (Menon, Gul & Sarnak, 2005; Choi et al., 2019; Cozzolino et al., 2018; Allawi, 2018). Among dialysis patients, magnesium deficiency has been shown to correlate with atherosclerosis and calcification of VSMC (Salem et al., 2012; Montes de Oca et al., 2014). A previous meta-analysis concluded that hypomagnesemia is significantly associated with all-cause and cardiovascular mortality in patients with chronic kidney disease (CKD) and ESRD, though the analysis did not analyze the outcomes for these two populations separately (Xiong et al., 2019). Another meta-analysis demonstrated that hypomagnesemia was closely related to an increase in all-cause mortality in hemodialysis (HD) patients, but the impact of serum magnesium on peritoneal dialysis (PD) patients was not discussed. This meta-analysis also did not investigate the diverse cut-off values of serum magnesium among the included studies (Liu & Wang, 2021). Because of these existing gaps in the published research, we conducted a meta-analysis about the association of serum magnesium with all-cause mortality and CV mortality in patients with ESRD on KRT.

Strengths and limitations

Our study has some significant strengths. This is the first meta-analysis comprehensively evaluating the association of serum magnesium concentration on all-cause mortality and CV mortality in dialysis patients, specifically. Our findings represent the current evidence supporting the potential influence of hypomagnesemia on relevant clinical outcomes. We also extracted data from the KM survival curve using the WebPlotDigitizer, making our data more complete.

However, our study results should be interpreted cautiously because of its limitations. First, a high heterogeneity was observed among the included studies, which could be related to differences among the studies in serum magnesium cut-off values and follow-up period lengths. Two included studies (Vervloet et al., 2013; Garagarza et al., 2015) were only available as abstracts and were thus lacking complete, available data. The inclusion criteria for patients also differed among the included studies. For example, the proportion of diabetes was vastly different among the studies, with one study even excluding DM patients (Shimohata et al., 2019). These inclusion differences mean the patients included in this meta-analysis may not match the characteristics of real-world patients on dialysis. Two of the included studies (Markaki et al., 2012; Vervloet et al., 2013) combined the data for both HD and PD patients, which may interfere with the analyses of HD and PD patients, separately. More studies investigating the relationship between serum magnesium and clinical outcomes in PD patients, specifically, are needed. In three of the included studies (Kurita et al., 2015; Lacson et al., 2015; Yang et al., 2016), participants were separated into multiple serum magnesium subgroups (ex. low, middle, higher groups). For these studies, we compared the outcomes of the lowest and the highest magnesium groups, which may have induced relevant bias in the effect estimates.

Conclusions

In our meta-regression analysis, the quantitative measures of serum magnesium levels, age, sex, or prevalence of diabetes were not associated with all-cause mortality or CV mortality. Our study suggested that a lower magnesium concentration is associated with a significant risk of all-cause mortality and CV mortality compared with non-hypomagnesemia serum magnesium levels in dialysis participants. Hypomagnesemia had a larger impact on the clinical outcomes of HD patients than for PD patients, but additional studies investigating serum magnesium levels in PD patients are required.

Supplemental Information

Supplemental Information 1 Search equation via PubMed, EMBASE, and Cochrane library.

Click here for additional data file.

Supplemental Information 2 PROSPERO protocol registration.

Click here for additional data file.

Supplemental Information 3 Newcastle-Ottawa Scale Quality Assessment of included studies.

Click here for additional data file.

Supplemental Information 4 PRISMA checklist.

Click here for additional data file.

Supplemental Information 5 Quality assessment the GRADE results.

Click here for additional data file.

Supplemental Information 6 Meta-regression analysis.

Click here for additional data file.

Supplemental Information 7 The funnel plot showing the visual check for publication bias of the effect of hypomagnesium compared with non-hypomagnesemia on the risk of all-cause mortality by pooling the hazard ratios.

Click here for additional data file.

Supplemental Information 8 The funnel plot showing the visual check for publication bias of the effect of hypomagnesium compared with non-hypomagnesemia on the risk of cardiovascular mortality by pooling the hazard ratios.

Click here for additional data file.

Supplemental Information 9 The meta-regression bubble plot showing the effect of the serum magnesium level of Age on all cause mortality.

Click here for additional data file.

Supplemental Information 10 The meta-regression bubble plot showing the effect of the serum magnesium level of DM on all-cause mortality.

Click here for additional data file.

Supplemental Information 11 The meta-regression bubble plot showing the effect of the serum magnesium level of (C) Male on all-cause mortality.

Click here for additional data file.

Supplemental Information 12 The meta-regression bubble plot showing the effect of the serum magnesium level of Age on CV mortality.

Click here for additional data file.

Supplemental Information 13 The meta-regression bubble plot showing the effect of the serum magnesium level of DM on CV mortality.

Click here for additional data file.

Supplemental Information 14 The meta-regression bubble plot showing the effect of the serum magnesium level of Male on CV mortality.

Click here for additional data file.

Supplemental Information 15 Rationale.

Click here for additional data file.

Supplemental Information 16 Data.

Click here for additional data file.

We thank for the staff of the division of the Nephrology, Internal medicine department of Chi Mei Medical Center, with valuable opinion for this study.

Additional Information and Declarations

Competing Interests

Author Contributions

Data Availability

The authors declare that they have no competing interests.

Chi-Ya Huang conceived and designed the experiments, prepared figures and/or tables, and approved the final draft.

Chi-Chen Yang conceived and designed the experiments, prepared figures and/or tables, and approved the final draft.

Kuo-Chuan Hung performed the experiments, authored or reviewed drafts of the article, and approved the final draft.

Ming-Yan Jiang analyzed the data, authored or reviewed drafts of the article, and approved the final draft.

Yun-Ting Huang performed the experiments, prepared figures and/or tables, and approved the final draft.

Jyh-Chang Hwang analyzed the data, authored or reviewed drafts of the article, and approved the final draft.

Chih-Chieh Hsieh analyzed the data, prepared figures and/or tables, and approved the final draft.

Min-Hsiang Chuang performed the experiments, prepared figures and/or tables, and approved the final draft.

Jui-Yi Chen conceived and designed the experiments, prepared figures and/or tables, and approved the final draft.

The following information was supplied regarding data availability:

The raw data is available in the Supplemental Files.

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
