# Peer review of "Association between hypomagnesemia and mortality among dialysis patients: a systematic review and meta-analysis"

_PeerJ, doi:10.7717/peerj.14203_

## Round 0.1 · original submission · Minor Revisions

A reorganizing of the Discussion and Introduction sections is preferable.

·

Basic reporting

The manuscript title is good and interesting to read the paper but in line 18 – The name of university is not right (Chia Nan not Nai) and the location is not written.

The introduction is great and good clue question for the study but
- I suggest to illustrate more in side effects of hypomagnesaemia in line 58 and serum Mg importance for dialysis patient to illustrate more the importance of the study.
- And in line 54 punctuation after (system), line 57 punctuation before reference as all the other references.

Experimental design

The Material and methods is good with clarification of structure of meta-analysis.
Inclusion and exclusion criteria is good explanation of resources however it didn’t explain the degree of agreement between the two investigator or it needed a third author or not.
Data extraction is good but I suggest if you clarify if there is a paper you find but can’t find more clarification for data extraction to support your study so you excluded it.
Quality assessment is important point so I suggest you clarify the degree of agreement between the two authors as it is an important point in the Meta analysis.
The outcomes and definition of hypomagnesaemia and non- hypomagnesaemia is great and organized.
Data synthesis and statistical analysis and subgroup analysis is ok
- The study search outcomes and included patients in line 136 you Saied that 4746 studies were identified from database and 224 studies were identified from registers then remove 1148 studies due to duplication then the remaining 3822 you exclude 3798 by title and abstract unlike in figure 1 could you please check the number?

- Study characteristics line 144,145,146 I suggest you put the reference in order in years as it will be easier to return to table 1 .In line 148 you put the study of Li et al.2015 (that is the right one but in table you but li et al study 2016 and they are two different study could you check for that please. In line 148 I suggest to make the study name as table ( vervloet 2013----broke 2013)

Validity of the findings

Statistic is good but please could you check
Table 1
Markaki (2012) age is 65±15

Figure 3 A
It is good illustration but I suggest you write random effects model above the figure
And below each figure I² is in percentage %
In table 2
Lu (2020) all-cause mortality HR (95% CI)—3.53(1.59 – 7.83)
And CV mortality HR (95% CI) ---4.28 (1.42 – 12.91)

·

Basic reporting

Accept

Experimental design

Accept

Validity of the findings

Accept

Additional comments

Accept

·

Basic reporting

I suppose a professional English edit will help with the readability of the manuscript.
The literature was well referenced and relevant.
The structure conforms to PeerJ standards and discipline norms.
Figures are relevant, high quality, well labelled & described.
Raw data supplied.

Experimental design

This is a secondary research within Scope of the journal.
Research question is well defined, relevant & meaningful.
Rigorous investigation was performed to a high technical & ethical standard.
Methods were described with almost sufficient detail & information to replicate.

Validity of the findings

All underlying data have been provided; they are robust, statistically sound, & controlled.
Conclusions are well stated, linked to the original research question & limited to supporting results. However, I think it may need some revisions.

Additional comments

General comments:
This is a well-designed meta-analysis on the Effect of hypomagnesemia on the outcome of patients undergoing dialysis (both PD and HD), that to the best of my knowledge been conducted for the first time. I think it is suitable for publication after conducting some revisions.
Specific Comments:
In line 54, what is (System)?
The point must come after citations, not before. For example:
Correct: ... (Simon et al.).
Wrong: ... .(Simon et al.)
It would benefit the manuscript if the first paragraph of the introduction be enriched with some Epidemiological facts.
I think the previous meta-analysis studies that the authors mentioned in the introduction fit better in the Discussion section. In my opinion, the introduction should mainly focus on the definitions of two conditions and give an insight to the readers if this issue I’d preventable or not. Reasons for hypomagnesia in ESRD are good to be mentioned here.
Did the authors check the search terms with the MeSH database? I also think that it would be better to mention search strategy in the Methods section.
Why did not the authors include case-control studies?
Was there a particular reason why the authors did not conduct metaregression based on the quantitative measures of serum levels of mg?
Although the authors reported results of metaregression of demographic characteristics of the patients, I think it would be beneficial if they mentioned these findings in tables (I mainly have comorbidities in mind).
I suggest the authors explain a little bit more about the following mechanism that they described in their work:
Fourth, patients with hypomagnesemia may have risk of electrolyte imbalance such as hypokalemia or hypocalcemia, which may cause cardiac arrhythmia(Hansen & Bruserud 2018).
I think it would be nice to discuss a bit the ways to prevent hypomagnesia-related mortality in dialysis patients.
I suggest authors mention their metaregression results in the conclusion section.

---

## Round 0.2 · accepted · Accept

As I found from the rebuttal letter and the revised version of the manuscript, all the reviewers' comments were satisfactorily resolved. The manuscript was significantly improved.

No further changes are needed.